# Randomised controlled trial on the effect of internet-delivered computerised cognitive–behavioural therapy on patients with insomnia who remain symptomatic following hypnotics: a study protocol

Daisuke Sato,[1,2] Naoki Yoshinaga,[3] Eiichi Nagai,[4] Hideki Hanaoka,[4] Yasunori Sato,[5] Eiji Shimizu[1,6,7]

For numbered affiliations see end of article.

**Correspondence to**
Daisuke Sato;
daisu.sato@gmail.com

## ABSTRACT

**Introduction** Insomnia has severe consequences for health. Primary care physicians in Japan commonly provide hypnotics, which is far from optimal. The recommended treatment for insomnia is cognitive–behavioural therapy (CBT). Access to trained therapists, however, is limited. Rather than face-to-face CBT, several researchers have studied internet-delivered computerised CBT (ICBT). This paper describes the study protocol for a randomised controlled trial (RCT) to evaluate effectiveness and feasibility of our newly developed five-step ICBT as an adjunct to usual care (UC) compared with UC alone for patients with insomnia who remain symptomatic following hypnotics.

**Methods and analysis** This proposed exploratory RCT comprises two parallel groups (ICBT+UC and UC alone) consisting of 15 participants each (n=30) diagnosed with insomnia who remain symptomatic after pharmacotherapy. We aim to evaluate the effectiveness of six intervention weeks. The primary outcome of insomnia severity will be the Pittsburgh Sleep Quality Index (PSQI) at week 6. Secondary outcomes include sleep onset latency, total sleep time, sleep efficiency extracted from PSQI, current feeling of refreshment and perceived soundness of sleep measured using visual analogue scale, number of awakenings, anxiety by Hospital Anxiety and Depression Scale, depression by Center for Epidemiologic Studies Depression Scale and quality of life by Euro Qol-5D. All measures will be assessed at weeks 0 (baseline), 6 (postintervention) and 12 (follow-up), and intention-to-treat analysis will be applied. The statistical analysis plan has been developed considering design of field materials.

**Ethics and dissemination** This study will be conducted at the academic outpatient clinic of Chiba University Hospital, Japan. Ethics approval was granted by the Institutional Review Board of Chiba University Hospital. All participants will be required to provide written informed consent. The trial will be implemented and reported in accordance with Consolidated Standards of Reporting Trials recommendations.

**Trial registration number** UMIN000021509; Pre-results.

## Strengths and limitations of this study

► This is the first randomised controlled trial to focus on internet-delivered computerised cognitive–behavioural therapy (ICBT) as a therapeutic option for insomnia patients who remain symptomatic following hypnotics.
► This study reflects good clinical practice, and its results will contribute to the development of second-line treatments and establish future treatment algorithms.
► A study limitation is the inability to elucidate specific effects of the ICBT programme, because a psychological placebo group to control for non-specific factors will not be employed and sleep estimates will not be based on objective measures such as polysomnography or actigraphy.

## INTRODUCTION

Insomnia is characterised by the inability to fall asleep or awakening too early in the morning or during the night, resulting in non-restorative sleep and decreased daytime functioning. Insomnia comprises a symptomatic state of difficulty in sleeping, related to significant daytime effects, experienced three or more times a week for over 3 months.[1] Worldwide epidemiological research reports the incidence of insomnia to be 10%–12%.[2–4] Although the incidence of insomnia is high, spontaneous improvement of the disorder is low. In one study, insomnia persisted a year later in 74% of participants with insomnia, and the disorder persisted for over 3 years in 46%.[5] The recently revised Diagnostic and Statistical Manual of Mental Disorders, Fifth Edition (DSM-5) underlines the need for clinical acknowledgement of sleep disorders.[1] This recommendation is supported by studies reporting that the incidence of mental and physical health comorbidities are high and

that insomnia is a factor associated with mood disorders,[6] cardiovascular disease[7] and hyperglycaemia.[8 9] From a population health perspective, sleep is a more important issue than was previously recognised.[10 11]

In general, insomnia is related to worsening fatigue, work efficiency and decreased quality of life (QOL).[12–14] In spite of this evidence of decreased productivity resulting from sleep deficits and it being a necessary diagnostic standard for insomnia, comparatively little research has been conducted on its association with QOL. Quite amazingly while insomnia's perceived sleep deficit is few, many people begin asking for help, particularly when insomnia's perceived difficulty on personal functioning such as daytime fatigue become more pronounced.[15 16] In epidemiological research, 20% of the most frequently cited reasons for visiting a sleep disorder clinic were daytime symptoms of fatigue, psychological suffering, physical discomfort as heartburn, a backflow of acid and food from the stomach into the oesophagus after eating and impaired work efficiency.[16] Moreover, studies performed by sleep clinicians at consultations show that patients with insomnia complain of disturbances of mood and cognitive abilities coexisting with severe levels of anxiousness, fatigue, pain and physical discomfort. Once QOL has been severely impacted, individuals often request a consultation.[17 18]

### Pharmacotherapy for insomnia

Although the causes of insomnia are at present inadequately understood, some of the most common treatments provided by primary physicians are drugs including benzodiazepines, non-benzodiazepines, orexin receptor antagonists, melatonin receptor agonists and antidepressants.[19] Furthermore, international research shows that physicians in Japan prescribe excess doses of benzodiazepine.[20] Unfortunately, pharmacotherapy is associated with a high incidence of adverse effects, and examples of side effects include daytime sleepiness and recurrent insomnia.[21] For this reason, pharmacotherapy is not recommended.[21] In recent years, two alternate types of intervention for patients with insomnia have been experimentally examined. The first is cognitive–behavioural therapy (CBT). CBT has similar or even better and more long-lasting outcomes than pharmacological therapy and lacks the negative side effects of pharmacological treatments.[22] Clinical practice guidelines suggest CBT, rather than pharmacotherapy, as the initial therapy for patients with insomnia.[23–25] The second type of intervention is chronobiological treatment, which enhances regular input into the body's circadian rhythms using bright light.[26] It has been reported that compared with CBT, the effect of light therapy on sleep maintenance is lower.[27] Vallières et al[28] reported that pharmacotherapy prior to the initiation of CBT appears to be less effective than the combined treatment of pharmacotherapy and CBT, followed by CBT alone. This study also found that early introduction of CBT contributes to maximising the effect of pharmacotherapy.

### CBT for insomnia

CBT, regarded as the first-choice treatment for persistent sleep deficits,[29 30] is a psychological treatment aiming to correct negative thought patterns and behaviours that influence the persistence of insomnia. CBT is composed of a variety of methods including a behavioural component (sleep restriction therapy, stimulus control therapy and relaxation) in combination with cognitive (managing anxiety about sleep and intrusive thoughts) and educational (sleep hygiene education) components. Sleep restriction therapy is an approach designed to reduce the time in bed to the actual time of sleep being achieved.[31] Stimulus control treatment is behavioural directions to associate the bed with sleep and to restore a consistent sleep rhythm.[32] Sleep hygiene education includes health practices such as physical exercise and environmental correlates such as light that may further or disturb sleep.[33] Studies examining the effect of CBT in patients with insomnia show that CBT has moderate-to-large effects on sleep efficiency and quality, latency to sleep onset and time of awakening.[24 34 35] Furthermore, a systemic review concluded that there is low-grade to moderate-grade evidence suggesting that CBT for insomnia is more effective than medication and that approximately 60% of patients receiving CBT respond to therapy and 39% enter remission.[36] Further research examining the effect of CBT on the daytime symptoms of patients with insomnia is needed. There is some preliminary evidence that CBT may provide generalised benefits,[37–39] as well as some exploratory data from small samples suggesting that CBT for patients with insomnia may reduce anxiety and depression symptoms.[40 41] However, a sufficiently powered, well-planned trial examining the relationship between characteristics of sleep, such as sleep quality and QOL, is required. Okajima et al[42] showed that CBT with behavioural analysis is more effective for insomnia and comorbid depressive symptoms than conventional treatment and that it promotes an earlier reduction in daily doses of hypnotic medication in patients with chronic insomnia who are resistant to pharmacological treatment.

### Internet-delivered computerised CBT (ICBT) for insomnia

Recent studies have administered CBT for insomnia as an internet-delivered computerised programme.[43] The choice of a web application is generally motivated by the observation that online programmes are more accessible, have low cost and can be completed at one's own time. They are also generally preferred over face-to-face interventions because they can be accessed anonymously, are convenient and can be used anywhere. In a recent trial, van Straten et al[44] showed statistically significant effects on sleep quality (Pittsburgh Sleep Quality Index (PSQI)), total sleep time and sleep efficiency in participants receiving ICBT for insomnia compared with a wait-list control group. Furthermore, ICBT for hypnotics-resistant insomnia was significantly more effective than the control treatment in reducing insomnia severity (Insomnia Severity Index), use of hypnotics and improving sleep efficiency.[45]

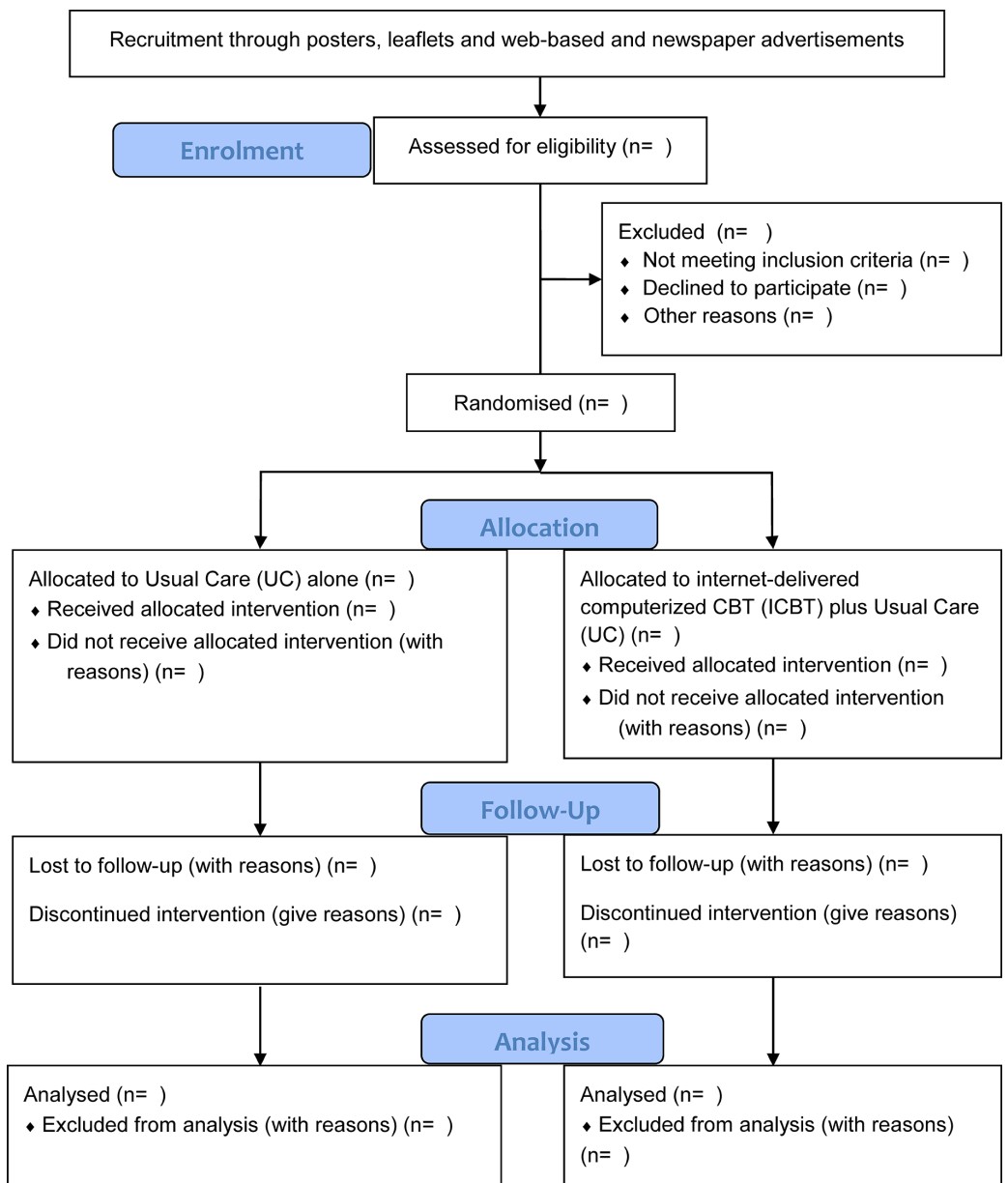

**Figure 1** CONSORT patients' flow diagram during inclusion, randomisation and treatment. CONSORT, Consolidated Standards of Reporting Trials.

## Aims

In summary, this paper describes the study protocol for a randomised controlled trial designed to evaluate the clinical effectiveness of ICBT adjunct to usual care (UC) for patients with insomnia who remain symptomatic after pharmacotherapy, compared with UC alone.[46 47]

## METHODS AND ANALYSIS
### Study design

This study was designed as a prospective randomised open-labelled endpoint trial ('PROBE') with two parallel intervention groups consisting of[1] a 6-week treatment regime of UC alone and[2] an ICBT combined with UC (figure 1).

## Participants

Inclusion criteria for this study include going to bed between 20:00 and 02:00, aged 18–65 years, having a primary diagnosis of insomnia according to the DSM-5 and the aforementioned insomnia remaining symptomatic. 'Remaining symptomatic' is defined herein as having insomnia that is at least moderate in severity, based on a PSQI score of >5.5, after the use of hypnotics including non-benzodiazepines, benzodiazepines, melatonin receptor agonists, orexin receptor antagonists and antidepressants.

Exclusion criteria include severe symptoms of anxiety or depression. Anxiety will be assessed using the anxiety subscale of the Hospital Anxiety and Depression Scale (HADS), which contains seven items. Depression will be

assessed using the total score of the Center for Epidemiological Studies Depression scale (CES-D). Participants with a HADS score of ≥10 or a CES-D score of ≥30 will be excluded. Participants with psychosis, organic mental disorder, or current high risk of suicide, substance abuse or dependence within the 12 months prior to enrolment, antisocial personality disorder or unstable medical condition will also be excluded.

## Eligibility procedure for participation and diagnosis

Treatment history will be confirmed by the prescribing clinician and by chart review. All participants will be evaluated by two researchers (a psychiatrist (ES) and a therapist (DS)) who will also verify patient diagnosis and eligibility. Validity of the initial diagnosis and eligibility will be discussed. Participants will be re-evaluated to cover important missing information based on suggestions derived from the discussion, and final diagnosis and eligibility will be confirmed by the two researchers.

## Recruitment

The planned recruitment rate is two participants per month, between March 2016 and January 2018, or until a total of 30 participants are recruited through posters and leaflets placed at medical institutions in Chiba Prefecture and through web-based and newspaper advertisements. As all participants will continue to be treated by their general practitioners, participants will be required to obtain permission from their general practitioners prior to study enrolment. This study will be conducted at the academic outpatient clinic of the Cognitive Behavioral Therapy Center of Chiba University Hospital, Japan.

## Interventions

### UC (conventional treatment)

Participants will be permitted to continue using hypnotics during the study period. There will be no restrictions on treatment options for participants who receive treatments, including medication changes, from their general practitioners. General practitioners will be permitted to refer participants for counselling or to secondary care if deemed clinically appropriate. However, the independent initiation of a strictly structured ICBT programme will be prohibited in order to properly assess the effectiveness of the current ICBT. All changes in conventional treatment, along with the reasons for those changes, will be recorded throughout the study period. As part of UC, both UC only and ICBT plus UC groups will receive magazines (in PDF format) developed and sent by email by our research team. Magazines will be sent four times over a 6-week period and will contain information about insomnia and hypnotics that are generally provided by general practitioners. Magazine content will include the following: (1) 'What you need to know about insomnia' (week 1); (2) 'What you need to know about sleeping drugs' (week 3); (3) 'How to use sleeping drugs correctly' (week 5); and (4) 'How to get off sleeping drugs in the near future' (week 6).

### ICBT programme

The ICBT programme for insomnia was developed by one of the researchers (ES). Treatment consists of 5 weekly lessons and includes various elements that are commonly incorporated in face-to-face CBT for insomnia. The primary components of this protocol include the following: (1) recording a sleep diary and understanding sleep hygiene; (2) changing sleep-related behaviours, including stimulus control; (3) restructuring distorted beliefs about sleep and sleep-related worries; (4) sleep restriction to increase sleep efficiency; and (5) relaxation training, including breathing exercises and progressive muscle relaxation. Participants will complete the five lessons over a 6-week period to provide sufficient time to become accustomed to CBT.

Every lesson will provide information, examples of other people carrying out the CBT treatment and homework. One of the researchers (DS) is a cognitive behavioural therapist and will send weekly emails to participants to ask them about their progress. The aim of this email support by a therapist is to allow participants to comment on the ICBT exercise, clarify information and motivate them to persist in carrying out the ICBT programme and requested homework. Participants will also be permitted to send emails to DS if they have questions about the content provided. Participants in the control group will also be offered the ICBT treatment after the trial, if the UC will not have made them sleep better.

## Outcomes

### Baseline and clinical characteristics

Baseline characteristics will include gender, age, education, marital status, employment status, age at insomnia onset and duration of insomnia. Moreover, treatment history will include the names of medications to which the participant has exhibited resistance, other prior treatments, current treatment (medication and others) at baseline and all changes in conventional treatment throughout the study period.

### Primary outcome

The primary outcome will be PSQI score at week 6. PSQI is a self-rated questionnaire consisting of 19 questions across seven subscales (sleep quality, sleep latency, sleep duration, habitual sleep efficiency, sleep disturbance, use of hypnotics and daytime dysfunction). Each subscale is scored on a scale of 0–3. Subscale scores are summed to a total score ranging from 0 (good quality of sleep) to 21 (very poor quality of sleep). PSQI was verified as a reliable and valid measure of subjective sleep quality in clinical practice and experimental research.[48 49]

### Secondary outcomes

Secondary outcomes will consist of sleep onset latency, total sleep time and sleep efficiency extracted from PSQI, as well as current feeling of refreshment and perceived soundness of sleep (assessed by visual analogue scale), number of awakenings (assessed by HADS) to measure

anxiety, CES-D score to measure depression and Euro Qol-5D (EQ-5D) score to measure QOL.

Total score on the seven HADS anxiety subscale items ranges from 0 (no symptoms of anxiety) to 21 (severe symptoms of anxiety). Total score on the 20 CES-D items ranges from 0 (no symptoms of depression) to 60 (severe symptoms of depression). The three-level version of EQ-5D was described in our previous study.[50] The EQ-5D[50] contains five items that assess QOL on a 3-point Likert scale ranging from 1 (not severe) to 3 (severe). The Japanese version of EQ-5D was developed by Tsuchiya *et al*.[51] EQ-5D is the most commonly used scale worldwide for calculating quality-adjusted life years (QALYs). QALYs are often used as the health outcome in cost–utility analyses and are typically estimated via area under the curve analysis, which involves summing the areas of the distribution shapes to calculate utility scores over the study period.[50 52] In the present study, QALYs will be assessed using EQ-5D, an indicator of patient health status. This index is calculated by transforming the EQ-5D dimension scores into a single summary score ranging from 0 to 1 (1=full health) by applying a formula created by the EuroQol Group.[50 53] Participants will complete these questionnaires at home.

The therapist will ask participants about adverse event experiences at each assessment. All measures will be assessed at weeks 0 (baseline), 6 (postintervention) and 12 (follow-up), and outcomes will be analysed based on the intention-to-treat principle.

## Sample size
The projected sample size was based on a previous study by van Straten *et al*,[44] which indicated that the estimated group difference in changes of PSQI scores from baseline was approximately 2.86 (ICBT group=3.00; control group=0.04). The conventional treatment (ie, UC) alone is assumed to be largely ineffective. Assuming a group difference of 2.86 points (SD=2.5), 13 subjects per group will provide 80% power to detect a difference in PSQI scores between the UC arm and ICBT plus UC arm for at least 6 weeks, using a two-sided, two-sample t-test at a 5% significance level. Thus, allowing for a 10% dropout rate, 15 participants are required per group, for a total of 30 participants in the study.

## Randomisation
At the end of the baseline assessment, eligible participants will be randomly assigned to either the UC arm or ICBT plus US arm at a ratio of 1:1, with assignments made using the minimisation method, ensuring a balance in baseline PSQI scores (PSQI ≥12) and gender.

Each participant will then be assigned to one of the two treatment regimes. Participants will be blinded to the group to which they are assigned before consenting to participate in the study.

## Data analysis plan
Statistical analysis and reporting of this trial will be conducted in accordance with Consolidated Standards of Reporting Trials (CONSORT) guidelines, with primary analyses based on the intention-to-treat principle. For baseline variables, summary statistics will be constructed, employing frequencies and proportions for categorical data and mean and SD for continuous variables. Baseline variables will be compared using the Fisher's exact test for categorical outcomes and the unpaired t-test for continuous variables.

For the primary analysis comparing treatment effects, the least-squares means and their 95% CIs will be estimated by analysis of covariance (ANCOVA) with the change in total PSQI scores at week 6. This ANCOVA model will take into account the variation caused by treatment effects, and gender and baseline PSQI score will be entered as covariates. Analyses of secondary outcomes will be performed in the same manner as the primary analysis.

All comparisons are planned, and all P values will be two-sided. P values<0.05 will be considered statistically significant. All statistical analyses will be performed using SAS V.9.4.

## Ethics and dissemination
This study will be conducted at the Academic Outpatient Clinic of the Cognitive Behavioral Therapy Center at Chiba University Hospital, Japan.

When potential participants contact the study trial office, they will be informed of the study objectives and asked if they are willing to participate. Each patient will be informed that participation is voluntary and that full anonymity will be provided. Each participant will then be required to provide written informed consent for their participation in this study. Each participant will also be informed that all participants will receive UC (conventional treatment) from their general practitioner and that half of the recruited participants will also receive ICBT in addition to their UC. A physician's examination will be conducted at each assessment point (weeks 0, 6 and 12).

An adverse event could consist of any unfavourable and unintended sign, symptom or disease temporally associated with this interventional study, whether considered related to the ICBT programme. All adverse events will be reported, and serious adverse events will be immediately reported to the Institutional Review Board of Chiba University Hospital in addition to being registered with the hospital risk management system. Moreover, an independent data monitoring committee will accurately verify detailed records of the clinical study's progress, critical efficacy variables and safety data, and recommend the sponsor to continue, modify or terminate the study accordingly.

Study outcomes will be published in international journals, regardless of the outcome. The study will be conducted and reported according to the CONSORT recommendations.

## DISCUSSION

This study was planned to address the lack of trials examining the administration of ICBT in patients with insomnia after hypnotic treatment. The findings of this study will provide valuable evidence to facilitate the development of second-line treatments after pharmacotherapy and to establish treatment algorithms. The foreseen limitations of this study are the following. First, sleep estimates will be based on subjective sleep diaries and PSQI scores, rather than on objective measures such as polysomnography or actigraphy. The use of both subjective and objective measures has been recommended because patients with insomnia underestimate their actual sleep time.[11 17] However, using polysomnography is costly and imposes significant burden on participants. Therefore, sleep diaries are currently the most widely used outcome measure in insomnia treatment studies[5] and are generally well accepted because subjective complaints generally prompt participants to seek treatment. The second limitation is that we will be unable to elucidate specific effects of the ICBT programme because a psychological placebo group to control for non-specific factors will not be employed. The third limitation is that there is a possibility for improvement of control group as ICBT is provided to the applicant of the control group after the end of the follow-up period.

**Author affiliations**
[1]Departments of Cognitive Behavioral Physiology, Chiba University Graduate School of Medicine, Chiba, Japan
[2]Department of Rehabilitation Sciences, Faculty of Health Care Sciences, Chiba Prefectural University of Health Sciences, Chiba, Japan
[3]Organization for Promotion of Tenure Track, University of Miyazaki, Miyazaki, Japan
[4]Clinical Research Center, Chiba University Hospital, Chiba, Japan
[5]Department of Global Clinical Research, Graduate School of Medicine, Chiba University, Chiba, Japan
[6]Cognitive Behavioral Therapy Center, Chiba University Hospital, Chiba, Japan
[7]Research Center for Child Mental Development, Chiba University, Chiba, Japan

**Contributors**  DS and ES especially made contribution to the design of this study, development of the original study protocol and drafting the initial manuscript. YS contributed to develop the statistical analysis plan and assisted in the preparation of the manuscript. NY, EN and HH contributed to the conceptualisation and design of this study and the critical revision of the article for important intellectual content. All authors approved the final version of the manuscript and agree to be accountable for all aspects of the work in ensuring that questions related to the accuracy or integrity of any part of the work are appropriately resolved.

**Funding**  This research received no specific grant from any funding agency in the public, commercial or not-for-profit sectors.

**Competing interests**  None declared.

**Patient consent**  Obtained.

**Ethics approval**  The study protocol has been approved by the Institutional Review Board of the Chiba University Hospital (reference number: G27040) on December 2015.

**Provenance and peer review**  Not commissioned; externally peer reviewed.

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
