## [Reviewer comments · BMJ Open]

ARTICLE DETAILS

TITLE (PROVISIONAL)	Study protocol for a randomized controlled trial on the effect of internet-delivered computerized cognitive behavioral therapy on patients with insomnia who remain symptomatic following hypnotics
AUTHORS	Sato, Daisuke; Yoshinaga, Naoki; Nagai, Eichi; Hanaoka, Hideki; Sato, Yasunori; Shimizu, Eiji

VERSION 1 – REVIEW

REVIEWER	Yoshio Minabe Dept of Psychiatry Kanazawa Univ School of Med, Ishikawa, Japan
REVIEW RETURNED	17-Aug-2017

GENERAL COMMENTS	The authors designed a pilot study for a randomized controlled trial to evaluate effectiveness and feasibility of our newly developed 5-step internet-delivered computerized cognitive behavioral therapy (ICBT) as an adjunct to usual care (UC) compared with UC alone for patients with insomnia who remain symptomatic following hypnotics. The protocol was well-written. The following points should be criticized. 1, In the introduction, the authors should mention that researchers in clinical practice guidelines of the American Academy of Sleep Medicine [Schutte-Rodin et al. 2008], the British Association for Psychopharmacology [Wilson et al., 2010], and the American College of Physicians [Qaseem et al., 2016] suggested cognitive behavioral therapy for insomnia (CBT-I) as the initial therapy rather than pharmacotherapy. Schutte-Rodin S, Broch L, Buysse D, et al. Clinical guideline for the evaluation and management of chronic insomnia in adults. J Clin Sleep Med 2008; 4:487. Wilson SJ, Nutt DJ, Alford C, et al. British Association for Psychopharmacology consensus statement on evidence-based treatment of insomnia, parasomnias and circadian rhythm disorders. J Psychopharmacol 2010; 24:1577. Qaseem A, Kansagara D, Forcica MA, et al. Management of Chronic Insomnia Disorder in Adults: A Clinical Practice Guideline From the American College of Physicians. Ann Intern Med 2016; 165:125. 2, Although the authors mentioned chronobiological treatment to enhance regular input into the body's circadian rhythms by use of bright light, the evidence is not as strong as CBT. In the introduction, the authors should describe effectiveness of light therapy, compared with CBT or pharmacotherapy.
---

	3. An exploratory study by Vallières et al. (2005) suggested that pharmacotherapy prior to the initiation of CBT appears to be less effective. The authors should discuss sequential combinations of drug and CBT. Vallières A, Morin CM, Guay B. Sequential combinations of drug and cognitive behavioral therapy for chronic insomnia: an exploratory study. Behav Res Ther. 2005;43(12):1611.
--	--

REVIEWER	Tanja van der Zweerde Phd-student Department of Clinical Psychology VU University Amsterdam The Netherlands
REVIEW RETURNED	24-Aug-2017

GENERAL COMMENTS	Thank you for the opportunity to review your article. I find it a very interesting topic, and your aim of offering better treatment to people suffering from insomnia is one I share with you. I have some comments that I hope you will find of use. Your protocol seems well thought-out and the article presents your design very concisely. A few points of consideration. 1) You mention chronobiological treatment as a treatment option that has been experimentally verified very briefly. I would suggest either elaborating some more, or alternatively leaving it out, as your main focus is on CBT. 2) I would like to see a definition of sleep-medication resistance, which makes clearer to the reader who your intended participants are. "Remaining symptomatic" should be defined more clearly. In addition, I think more information is needed on what treatment history the intended participants should have had (i.e. medication, but for how long, dosage, type of drug, several attempted sleep drugs, etc.). Are participants allowed to keep using medication while undergoing CBTI? 3) I have doubts about the power calculation. I believe in the study by Van Straten et al. (2014) the effects found were bigger than what can be expected in your trial, since Van Straten et al. used a wait-list control group that received no treatment. In your design, control group participants will be undergoing treatment as usual. Would they not be expected to have (albeit smaller) improvements in their sleep as well? 4) Sleep restriction therapy, sleep hygiene education and stimulus control treatment are mentioned but not explained. I believe these are not self-explanatory and deserve elaboration. 5) Will participants in the control group also be offered the CBTI treatment after the trial, if the usual care (as you expect) will not have made them sleep better? I think this would be advisable from a ethical viewpoint. I believe the paper could benefit from careful editing and proof reading from a English native speaker, as I came across a number of spelling/formulation mistakes. For example: 1. introduction line 24: word 'the' (mood disorder) should be removed, and -s added (mood disorders)
--

	2. introduction line 33: capital L (Little) should be small 'l'. 3. introduction line 34-37: this sentence is unclear and needs rephrasing ("Quite - sleep deficit"). 4. introduction line 41/42: "physical uncomfortable feeling" needs rephrasing 5. introduction line 46-48: this sentence is unclear and needs rephrasing ("Therefore - ask for consultation"). 6. introduction page 2 line 7/8: risk of addiction and abuse are not side effects of the drugs but unfavorable risks of its use 7. introduction page 2 line 10 "Recent years" should be "In recent years" 8. Several errors in spacing and interpunction 9. The references are not always correct (for example, at the end of the introduction you refer to (37) while Van Straten et al. is reference (38). Please check the references. I hope you will find my comments of use and I wish you the best of luck with performing this interesting study.
--	--

VERSION 1 – AUTHOR RESPONSE

Response to Comments from Reviewer 1

Comment #1-1

In the introduction, the authors should mention that researchers in clinical practice guidelines of the American Academy of Sleep Medicine [Schutte-Rodin et al. 2008], the British Association for Psychopharmacology [Wilson et al., 2010], and the American College of Physicians [Qaseem et al., 2016] suggested cognitive behavioral therapy for insomnia (CBT-I) as the initial therapy rather than pharmacotherapy.

Response

Thank you for your detailed comment. We have added the description in Introduction (Pharmacotherapy for insomnia), "Clinical practice guidelines suggest CBT, rather than pharmacotherapy, as the initial therapy for patients with insomnia."

Comment #1-2

Although the authors mentioned chronobiological treatment to enhance regular input into the body's circadian rhythms by use of bright light, the evidence is not as strong as CBT. In the introduction, the authors should describe effectiveness of light therapy, compared with CBT or pharmacotherapy.

Response

We have added the description in Introduction (Pharmacotherapy for insomnia), "It has been reported that compared with CBT, the effect of light therapy on sleep maintenance is lower."

Comment #1-3

An exploratory study by Vallières et al. (2005) suggested that pharmacotherapy prior to the initiation of CBT appears to be less effective. The authors should discuss sequential combinations of drug and CBT.

Response

We have added the description in Introduction (Pharmacotherapy for insomnia), "Vallières reported that pharmacotherapy prior to the initiation of CBT appears to be less effective than the combined

treatment of pharmacotherapy and CBT, followed by CBT alone. This study also found that early introduction of CBT contributes to maximizing the effect of pharmacotherapy.”.

Response to Comments from Reviewer 2

Comment #2-1

You mention chronobiological treatment as a treatment option that has been experimentally verified very briefly. I would suggest either elaborating some more, or alternatively leaving it out, as your main focus is on CBT.

Response

We have added the description in Introduction (Pharmacotherapy for insomnia), “It has been reported that compared with CBT, the effect of light therapy on sleep maintenance is lower.”.

Comment #2-2

I would like to see a definition of sleep-medication resistance, which makes clearer to the reader who your intended participants are.

“Remaining symptomatic” should be defined more clearly. In addition, I think more information is needed on what treatment history the intended participants should have had (i.e. medication, but for how long, dosage, type of drug, several attempted sleep drugs, etc.). Are participants allowed to keep using medication while undergoing CBTI?

Response

We have modified the description in Methods and Analysis (Participants), ““Remaining symptomatic” is defined herein as having insomnia that is at least moderate in severity, based on a PSQI score of >5.5, after the use of hypnotics including non-benzodiazepines, benzodiazepines, melatonin receptor agonists, orexin receptor antagonists, and antidepressants.”.

Participants are allowed to keep using medication during this study period”. We have changed the word in Methods and Analysis (Interventions, UC), “hypnotics” instead of “medication”.

Comment #2-3

I have doubts about the power calculation. I believe in the study by Van Straten et al. (2014) the effects found were bigger than what can be expected in your trial, since Van Straten et al. used a wait-list control group that received no treatment. In your design, control group participants will be undergoing treatment as usual.

Would they not be expected to have (albeit smaller) improvements in their sleep as well?

Response

That is as you pointed out. There is a possibility for improvement of control group. We have added additional comments on the limitations of the study in Discussion, “The third limitation is that there is a possibility for improvement of control group as ICBT is provided to the applicant of the control group after the end of the follow-up period.”

Comment #2-4

Sleep restriction therapy, sleep hygiene education and stimulus control treatment are mentioned but not explained. I believe these are not self-explanatory and deserve elaboration.

Response

We have added the description in Introduction (CBT for insomnia), “Sleep restriction therapy is an approach designed to reduce the time in bed to the actual time of sleep being achieved”, “Stimulus

control treatment is behavioral directions to associate the bed with sleep and to restore a consistent sleep rhythm”, and “Sleep hygiene education includes health practices such as physical exercise and environmental correlates such as light that may further or disturb sleep”.

Comment #2-5

Will participants in the control group also be offered the CBTI treatment after the trial, if the usual care (as you expect) will not have made them sleep better?

I think this would be advisable from an ethical viewpoint.

I believe the paper could benefit from careful editing and proof reading from an English native speaker, as I came across a number of spelling/formulation mistakes. For example:

1. introduction line 24: word 'the' (mood disorder) should be removed, and -s added (mood disorders)
2. introduction line 33: capital L (Little) should be small 'l'.
3. introduction line 34-37: this sentence is unclear and needs rephrasing ("Quite - sleep deficit").
4. introduction line 41/42: "physical uncomfortable feeling" needs rephrasing
5. introduction line 46-48: this sentence is unclear and needs rephrasing ("Therefore -ask for consultation").
6. introduction page 2 line 7/8: risk of addiction and abuse are not side effects of the drugs but unfavorable risks of its use
7. introduction page 2 line 10 "Recent years" should be "In recent years"
8. Several errors in spacing and interpunction
9. The references are not always correct (for example, at the end of the introduction you refer to (37) while Van Straten et al. is reference (38). Please check the references.

Response

We have added the description in Methods and Analysis (Interventions, ICBT program), "Participants in the control group will also be offered the ICBT treatment after the trial, if the usual care will not have made them sleep better."

1. introduction line 24: We have removed word 'the' (mood disorder), and added '-s' (mood disorders).
2. introduction line 33: We have changed small 'l' (little), instead of capital L (Little).
3. introduction line 34-37: We have modified the description in Introduction, "Quite amazingly while insomnia's perceived sleep deficit is few, many people begin asking for help, particularly when insomnia's perceived difficulty on personal functioning such as daytime fatigue become more pronounced."
4. introduction line 41/42: We have modified the description in Introduction, " physical uncomfortable feeling as heartburn, a backflow of acid and food from the stomach into the esophagus after eating".
5. introduction line 46-48: We have modified the description in Introduction, "Once QOL has been severely impacted, individuals often request a consultation."
6. introduction page 2 line 7/8: We have remove the sentence, "risk of addiction and abuse".
7. introduction page 2 line 10: We have changed to "In recent years", instead of "Recent years".
8. Several errors in spacing and interpunction: We have been proofread and edited by a native speaker again.
9. check the references: We have checked the references again for any mistakes.